# Emergency admissions and long-term conditions during transition from paediatric to adult care: a cross-sectional study using Hospital Episode Statistics data

Linda Petronella Martina Maria Wijlaars,[1,2] Pia Hardelid,[1] Astrid Guttmann,[3] Ruth Gilbert[1,2]

[1]Population, Policy and Practice/Children's Policy Research Unit, University College London Great Ormond Street Institute of Child Health, London, UK
[2]Administrative Data Research Centre for England, University College London, London, UK
[3]Health System Planning & Evaluation Research Program, Institute for Clinical Evaluative Sciences, Toronto, Ontario, Canada

**Correspondence to**
Dr Linda Petronella Martina Maria Wijlaars;
linda.wijlaars@ucl.ac.uk

## ABSTRACT

**Objective** To determine whether changes in emergency admission rates during transition from paediatric to adult hospital services differed in children and young people (CYP) with and without underlying long-term conditions (LTCs).

**Design** Cross-sectional study.

**Setting** Emergency admissions between 2009 and 2011 recorded in the Hospital Episode Statistics Admitted Patient Care data in England.

**Participants** 763 199 CYP aged 10–24 years with and without underlying LTCs (LTCs were defined using the International Classification of Diseases, 10th Revision codes recorded in the past 5 years).

**Primary and secondary outcome measures** We calculated emergency admission rates before (10–15 years) and after transition (19–24 years), stratified by gender, LTC and primary diagnosis. We used negative binomial regression to estimate adjusted incidence rate ratios (IRRs).

**Results** We included 1 109 978 emergency admissions, of which 63.2% were in children with LTCs. The emergency admission rate increased across the age of transition for all CYP, more so for those with LTCs ($IRR_{LTC}$: 1.55, 99% CI 1.47 to 1.63), compared with those without ($IRR_{noLTC}$: 1.21, 99% CI 1.18 to 1.23). The rates increased most rapidly for CYP with mental health problems, MEDReG (metabolic, endocrine, digestive, renal, genitourinary) disorders, and multiple LTCs (both genders) and respiratory disorders (female only). Small or no increased rates were found for CYP without LTCs and for those with cancer or cardiovascular disease. Increases in length of stay were driven by long admissions (10+ days) for a minority (1%) of CYP with mental health problems and potentially psychosomatic symptoms. Non-specific symptoms related to abdominal pain (girls only), gastrointestinal and respiratory problems were the most frequent primary diagnoses.

**Conclusions** The increased rates and duration of emergency admissions and predominance of non-specific admission diagnoses during transition in CYP with underlying LTCs may reflect unmet physical or mental health needs.

### Strengths and limitations of this study

► This study included all hospital admissions to the National Health Service in England over 3 years, minimising selection bias and allowing for comparison of patients using paediatric and adult services.
► Our analysis used a standardised and validated coding system to identify underlying long-term conditions, based on any diagnostic codes recorded at the current or past admissions in the previous 5 years.
► As we used a cross-sectional design with admissions as the denominator, we did not account for individuals with multiple admissions (ie, clustering) and could have overestimated the contribution of certain conditions that are associated with multiple emergency admissions.
► We may have included young people who were first diagnosed with long-term conditions during or after the age of transition.

## BACKGROUND

Transition from paediatric to adult services presents challenges to children and young people (CYP), their families and to service providers.[1] First, the team managing the CYP's condition often shifts from specialist, hospital-based to generic, community-based care by general practitioners (GPs). For instance, for diabetes care in the UK, care for children is coordinated by paediatric endocrinologists in the hospital, while diabetes care for adults is led by primary care.[2] Additionally, as the clinicians and thresholds for access to specialist care change with the shift to adult services, ongoing healthcare is often disrupted. This can be particularly problematic for children with long-term conditions (LTCs), who account for 41% of all emergency admissions in children in England,[3]

as they often have little contact with their primary care physician prior to transition.[4]

Second, transition to adult care occurs during a critical and vulnerable period for children with LTCs. Adult services often expect CYP to assume responsibility for their own care despite the fact that psychological, social and physiological changes associated with adolescence can disrupt self-management of LTCs.[5 6] Finally, the types of clinical problems experienced by CYP during transition are changing due to the increased survival of young children with complex or multiple LTCs partly due to improved care and support for (very) preterm infants and for CYP with conditions that would previously have been lethal in early childhood.[7 8]

We have previously reported that rates of emergency and elective admissions increase in young people around the time of transition to adult services.[9] Most studies on care across the transition to adult care have used qualitative methods to evaluate patient or clinician expectations and experience,[10] or have studied outcomes in a relatively small, regionally defined or disease-specific population.[11 12] Our primary aim was to use cross-sectional analyses of hospital administrative data for England to compare changes in rates and duration of emergency admission to hospital for CYP with and without underlying LTCs before and after the period of transition from paediatric to adult care. In secondary analyses, we explored changes for LTC groups and determined which primary diagnostic groups accounted for changes in admission rates.

## METHODS

### Data source

We used anonymised patient-level Hospital Episode Statistics data for admissions to English National Health Service (NHS) hospitals from 1 April 2009 to 31 March 2012.[13] We used 3 years of data to ensure any patterns were not due to single-year fluctuations. We identified admissions as continuous periods in the hospital that could consist of several finished consultant episodes (a period of hospital stay under a single consultant).[14] Admissions that occurred within a day of each other or admissions that included a hospital transfer were considered as a single admission.

### Population

We included adolescents and young people aged 10–24 years in our analysis in order to determine incidence rates before (ages 10–15 years) and after (19–24 years) transition. We defined transition from paediatric to adult healthcare between the ages of 16 and 18 years, as in practice the exact age of transition in the NHS can vary between different NHS services and/or conditions.[15–18]

Our main outcome measure was the incidence rate ratio (IRR) of emergency admissions after compared with before the age of transition. We considered emergency (or unplanned) admission to be a clinically important

indicator of ill-health, which might be affected by the quality of healthcare received from hospital or care in the community during transition. We excluded emergency injury admissions from the primary outcome as these would be strongly affected by personal factors or public health or education interventions (eg, reaching the legal driving and legal drinking age).[9 19] However, injury admissions due to intentional self-harm (identified by a second diagnosis code indicating self-harm[20]) were included as these could signify an underlying mental health problem. We further excluded admissions related to pregnancy and maternities as we expected these to increase with age unrelated to transition from paediatric to adult services.

### Variable definitions

We identified CYP with and without underlying LTCs using validated *International Classification of Diseases, 10th Revision* (ICD-10) codes[14] recorded in any diagnostic code during the index admission or any hospital admission in the previous 5 years. We defined nine LTC groups: (1) mental health, (2) cancer and blood disorders, (3) chronic infections, (4) respiratory disorders, (5) metabolic, endocrine, digestive, renal, genitourinary disorders (MEDReG), (6) musculoskeletal and skin disorders, (7) neurological disorders, (8) cardiovascular disorders and (9) multiple LTCs (for children with LTCs from more than one group).

Other characteristics included as potential confounders were ethnicity, quintile of deprivation and whether the CYP was referred by a GP or emergency department. Deprivation was based on the postcode of residence at admission and measured using the Index of Multiple Deprivation 2004.[21] Analyses were stratified by sex.

We used the primary diagnosis at admission to define the main reason for admission, grouped as LTCs (only as primary diagnosis), infection, signs and symptoms, and other (see online supplementary appendix A, eTable A1). We calculated the length of stay by subtracting the discharge date from the admission date and classified admissions as day cases if admission and discharge occurred on the same date.

### Statistical analyses

As this is an exploratory study, we used a cross-sectional design. We calculated incidence rates for the years 2009–2011 per 100 person-years by sex, using the number of admissions for each calendar year and age group divided by the midyear population estimates as the denominator population from the Office for National Statistics.[22] The primary analysis determined the change in the incidence of emergency inpatient admissions measured by the IRR for the average incidence after transition (19–24 years) divided by the average incidence before transition (10–15 years).

We determined whether the IRR varied significantly according to whether a child had an underlying LTC recorded in the past 5 years, as well as by primary diagnosis at admission. To correct these estimates for confounding,

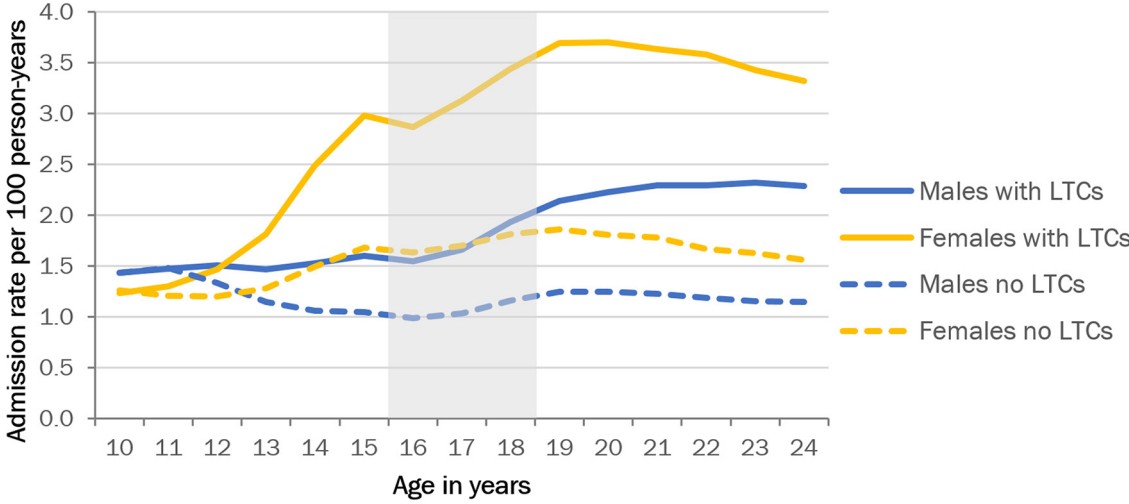

**Figure 1** Emergency and elective admission rates by age and sex. Long-term conditions (LTCs) as measured in the index admission and admissions in the previous 5 years. Grey area indicates age range for transition.

we used a negative binomial regression model (to correct for overdispersion) with admission counts as the outcome variable, age, underlying LTC, sex, deprivation quintile and ethnicity as explanatory variables, and population size as the offset. We used linear combination of regression coefficients to describe the IRRs for subgroups. To determine the effect of LTCs on the increase in admission rates across transition, we calculated the population-attributable fraction (PAF).[23]

We conducted sensitivity analyses to assess whether the results changed when including admissions related to injury and maternity. In addition, we assessed the effect of changing the criterion for underlying LTCs from being recorded in the previous 5 years to 3 or 1 years.

All analyses were stratified by sex and performed on Stata SE V.13.0. Figures were produced using Microsoft Excel.

### Patient and public involvement
Patients were not involved in the design of this study. In order to disseminate our results to young people, we worked with the National Children's Bureau's Young Research Advisors group to develop a short animation (https://youtu.be/u_NG0Vb3tec) that highlights the results they found most important and explore future research priorities.

## RESULTS
### Overall
We analysed 1 109 978 emergency admissions in 763 199 CYP aged 10–24 years. Of these, 680 322 (57.3%) admissions occurred in girls. Overall, 701 834 (63.2%) admissions occurred in CYP with a record of an LTC in the last 5 years (online supplementary eTable B1 in appendix B). This proportion increased from 57.2% in patients aged 10–15 years old to 66.6% in those aged 19–24 years old.

The increase in the total rate of emergency admissions (excluding elective admissions and emergency admissions for injury and pregnancy-related problems; online supplementary eFigure B1 in appendix B) was higher in girls than in boys: the IRRs comparing the average rates after with before transition were 1.64 (99% CI 1.63 to 1.64) for girls and 1.17 (99% CI 1.16 to 1.18) for boys. The increase in the rates of emergency admission in both sexes occurred predominantly in CYP with underlying LTCs (figure 1). In the fully adjusted model (table 1), transition was associated with a modest increase in emergency admission rates for children without LTCs (IRR: 1.21, 99% CI 1.18 to 1.23). However, for CYP with LTCs, the effect of transition was stronger (IRR: 1.55, 99% CI 1.47 to 1.63).

If CYP with LTCs had the same increase in the rates of emergency admission across transition as CYP without LTCs, the overall increase in admissions would decrease by 26.3% (PAF, 99% CI 25.9% to 26.7%) among girls and 25.7% (99% CI 25.2% to 26.1%) among boys.

Inequalities increased across transition (figure 2). Emergency admission rates varied from 2.2 to 3.6 per 100 person-years from the most affluent to the most deprived quintiles when children were aged 10 years, but this gap increased to 3.8 to 7.0 for those aged 24 years old.

### Admission rates by specific underlying LTC
The steepest increase in emergency admission rates occurred in CYP who had records indicating underlying mental health problems (figure 3, online supplementary eTable B2 in appendix B). Mental health problems were recorded in 31.2% of emergency admissions in those aged 16–18 years old, predominantly disorders related to alcohol/drug use, self-harm, depression, eating disorders and schizophrenia. CYP with records indicating MEDReG disorders or multiple LTCs also experienced large increases in admission rates across transition. Emergency admissions for cancer and blood disorders and cardiovascular conditions remained stable or decreased across transition.

**Table 1** Relative admission rates for CYP with and without LTCs and before (10–15 years) and after (19–24 years) transition

| Covariate | No LTC | LTC | No LTC | LTC |
|---|---|---|---|---|
| | Crude IRR (99% CI) | | Adjusted IRR (99% CI) | |
| Transition | 1.18 (1.17 to 1.18) | 1.44 (1.41 to 1.47) | 1.21 (1.18 to 1.23) | 1.55 (1.47 to 1.63) |
| Sex (reference=male) | 1.07 (1.06 to 1.07) | 0.92 (0.90 to 0.94) | 1.05 (1.05 to 1.06) | 0.98 (0.96 to 1.00) |
| Age | 1.02 (1.01 to 1.02) | 1.03 (1.03 to 1.04) | 1.00 (0.99 to 1.00) | 0.98 (0.98 to 0.99) |
| Index of Multiple Deprivation 2004 | | | | |
| 1—most deprived | 1.10 (1.09 to 1.11) | 1.41 (1.36 to 1.46) | 1.07 (1.05 to 1.08) | 1.17 (1.13 to 1.20) |
| 2 | 1.07 (1.06 to 1.09) | 1.28 (1.23 to 1.32) | 1.04 (1.03 to 1.06) | 1.08 (1.05 to 1.12) |
| 3 | 1.07 (1.06 to 1.10) | 1.05 (1.01 to 1.09) | 1.07 (1.05 to 1.08) | 1.02 (0.99 to 1.05) |
| 4 | 1.02 (1.01 to 1.04) | 1.04 (1.00 to 1.07) | 1.02 (1.00 to 1.03) | 1.02 (0.98 to 1.05) |
| 5—least deprived | Reference | Reference | Reference | Reference |
| *Missing* | 0.88 (0.86 to 0.91) | 1.11 (1.01 to 1.22) | 0.87 (0.85 to 0.89) | 1.01 (0.92 to 1.10) |
| Ethnicity | | | | |
| White | Reference | Reference | Reference | Reference |
| Black | 1.21 (1.18 to 1.24) | 2.67 (2.56 to 2.79) | 1.19 (1.16 to 1.22) | 2.49 (2.39 to 2.60) |
| Asian | 1.03 (1.01 to 1.04) | 1.07 (1.03 to 1.12) | 1.03 (1.01 to 1.04) | 1.13 (1.08 to 1.19) |
| Mixed | 0.94 (0.92 to 0.95) | 0.93 (0.89 to 0.97) | 0.93 (0.92 to 0.95) | 0.93 (0.89 to 0.97) |
| Unknown | 0.75 (0.74 to 0.77) | 0.43 (0.39 to 0.47) | 0.75 (0.74 to 0.77) | 0.43 (0.40 to 0.47) |
| *Missing* | 0.82 (0.80 to 0.84) | 0.52 (0.49 to 0.54) | 0.82 (0.81 to 0.84) | 0.54 (0.51 to 0.57) |

Age is included in the model as a continuous variable. IRR refers to increase per year of age.

Models are corrected for variables shown in the table.

The IRR compares the emergency admission rate before (age 10–15 years) transition with the rate after (age 19–24 years) transition.

CYP, children and young people; IRR, incidence rate ratio; LTC, long-term conditions.

Specific underlying LTCs of asthma, diabetes and epilepsy increased only marginally, although more so in girls (figures 3 and 4). As expected, larger increases were seen for inflammatory bowel disease (IBD), as this condition often emerges during adolescence. Admissions of CYP with these underlying LTCs (asthma, diabetes, epilepsy or IBD) contributed to a minority of admissions before (14.8%) and after (11.3%) transition. Emergency admissions for these four conditions accounted for 14.1%

(PAF, 99% CI 13.8% to 14.6%) of the increase across transition, just over half of the total effect (26.0%).

## Primary diagnosis on admissions

Although admissions increased in CYP with underlying LTCs, the primary reason for admission was often not the LTC itself. We found only modest increases in rates of emergency admission during transition with primary diagnoses recorded for specific LTCs (figure 3, online supplementary

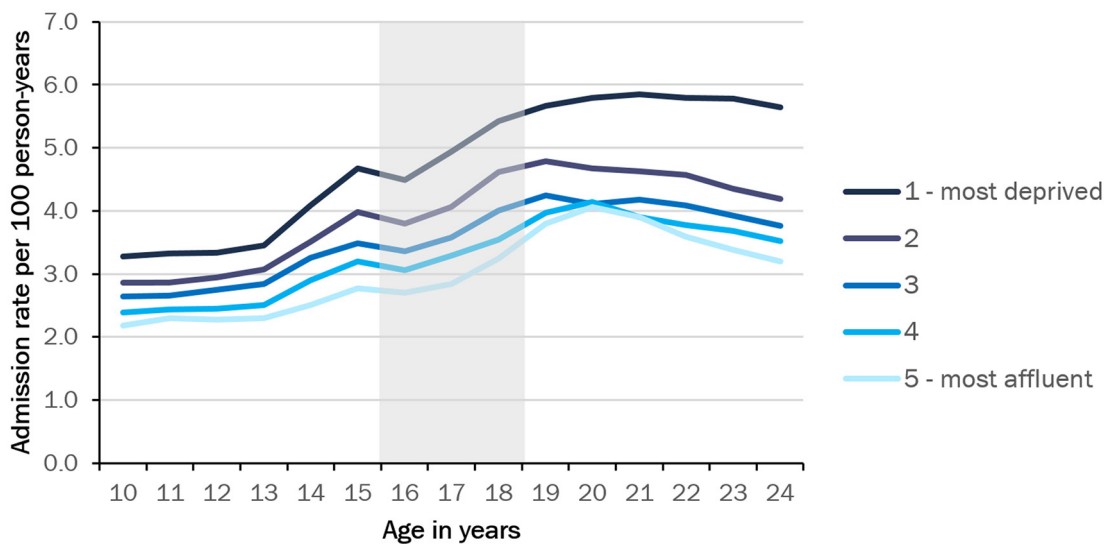

**Figure 2** Emergency admissions by Index of Multiple Deprivation (2004) quintile. Grey area indicates age range for transition.

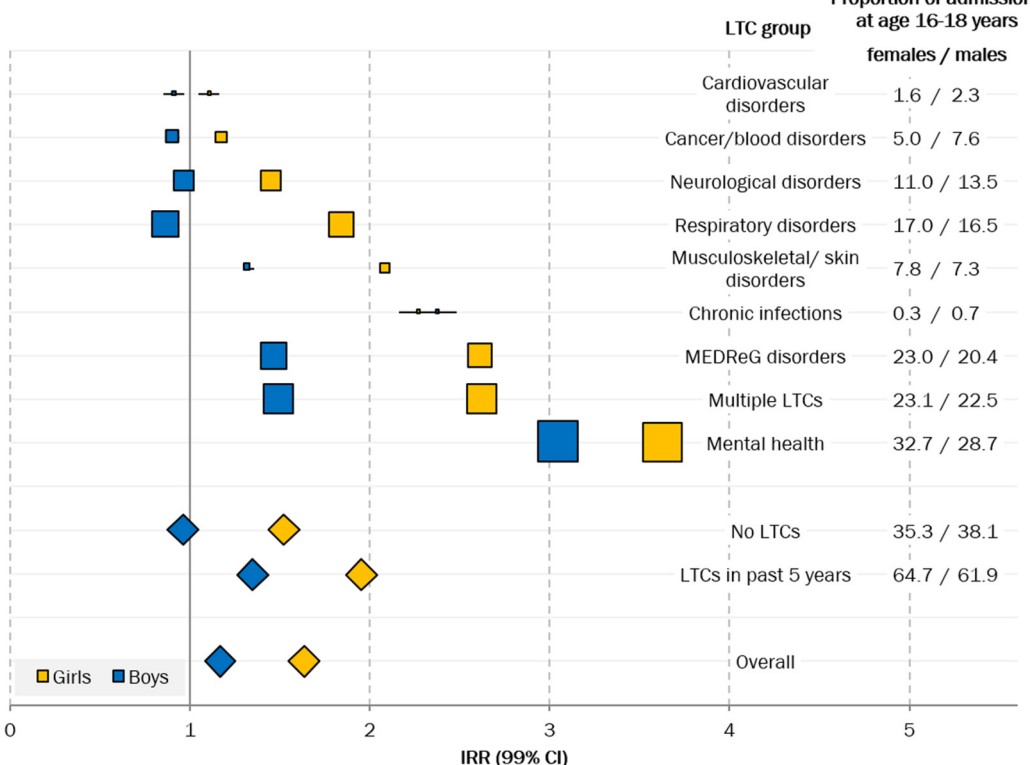

**Figure 3** IRRs for emergency admissions before (10–15 years) and after (19–24 years) transition by underlying long-term condition (LTC) groups and sex. Marker size represents the proportion of admissions during transition (age 16–18 years) with a record of LTCs (displayed on the right). Diamonds indicate group/overall estimates. IRR, incidence rate ratio; MEDReG disorders, metabolic, endocrine, digestive, renal, genitourinary disorders.

eTable B3 in appendix B). Admissions with a primary diagnosis of asthma or epilepsy remained relatively stable across transition (figure 4) while IBD increased.

As primary diagnoses for LTCs did not explain the increase in emergency admission rates (online supplementary eTable B4 in appendix B), we explored other reasons for admissions. Emergency admissions with primary diagnoses for 'signs and symptoms' saw strong increases over the transition. Admissions for abdominal pain, circulatory and respiratory symptoms, and general symptoms all increased substantially. Abdominal pain in particular was the primary reason for emergency admission in 19.7% (for girls) and 9.8% (for boys) at ages 16–18 years. Overall, emergency admission rates for symptoms increased for both girls (IRR: 1.83, 99% CI 1.81 to 1.84) and boys (IRR: 1.21, 99% CI 1.20 to 1.23) (online supplementary eTable B3 in appendix B).

Emergency admissions for infections also increased (IRR: 1.72 for girls, 1.29 for boys; online supplemetary eTable B3 in appendix B). There were also increases in 'any other' diagnoses for emergency admissions (where the primary diagnosis was not an LTC, infection nor symptom), although admission rates for individual reasons were low (online supplementary eTable B3 in appendix B).

### Duration of admission
The length of stay for emergency admissions increased with age from an average of 2.2 days at age 10 years to 4.5

days at age 24 years for girls, and from 1.9 to 9.2 for boys (figure 5). The median length of stay was 1 day for the entire age range, for both boys and girls. While day cases were the most common type of admission at age 10 years (38% for girls and 41% for boys), there was a drop in the proportion of day cases around the age of transition (age 16–18 years) to 27% for girls and 28% for boys.

A small proportion of girls and boys had very long admissions (10 days or longer, n=84812, 7.1% of the total emergency admissions), which increased from 5% before to 7% after transition for girls and from 4% to 11% for boys. A large proportion (18 812, 22.3%) of these long-stay admissions were related to mental health problems such as schizophrenia and personality disorders (online supplementary eTable B5 in appendix B), and a minority (4358, 5.2%) had an unspecified primary diagnosis (ICD-10 code R69.X).

Young adults were more likely to be admitted via the emergency department rather than via their primary care physician or other methods: the proportion of emergency admissions via the emergency department increased for girls from 51% before transition to 69% after transition and from 53% to 71% for boys.

### Sensitivity analyses
The results of the sensitivity analyses (online supplementary appendix C) show that by excluding injury and maternity/pregnancy-related admissions

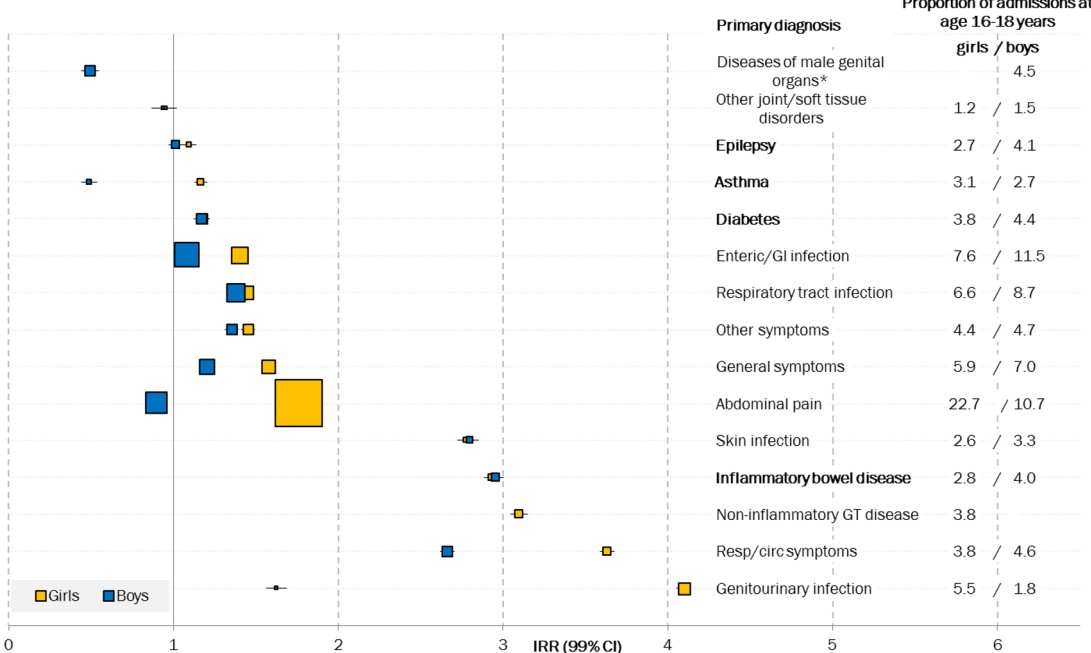

**Figure 4** IRRs for emergency admissions before (10–15 years) and after (19–24 years) transition by primary diagnosis and sex. Marker size represents the proportion of admissions at age 16–18 years with each primary diagnosis. The 15 primary diagnoses shown represent 76.9% and 77.7% of all primary diagnoses for girls and boys, respectively. LTCs are highlighted in bold. *Diseases of male genital organs (ICD-10 codes N40-N51) include torsion of testis, and inflammatory diseases of the male genital organs such as orchitis and epididymitis. GI, gastrointestinal; GT, genitourinary tract; ICD-10, International Classification of Diseases, 10th Revision; IRR, incidence rate ratio; LTC, long-term condition.

(online supplementary eTable C1), we have underestimated the overall increase in emergency admission rates for girls (26%) and boys (6%). However, the higher rate of admission in girls was entirely due to maternity-related emergency admissions, which increased from 0.4 to 22.2 per 100 person-years.

IRRs for emergency admissions before and after transition for CYP affected by underlying LTCs decreased slightly (from 1.88, 99% CI 1.87 to 1.89, to 1.83, 99% CI 1.82 to 1.84) when we changed our definition from LTCs recorded in the previous 5 years, to 3 or 1 year (online supplementary aTable C2).

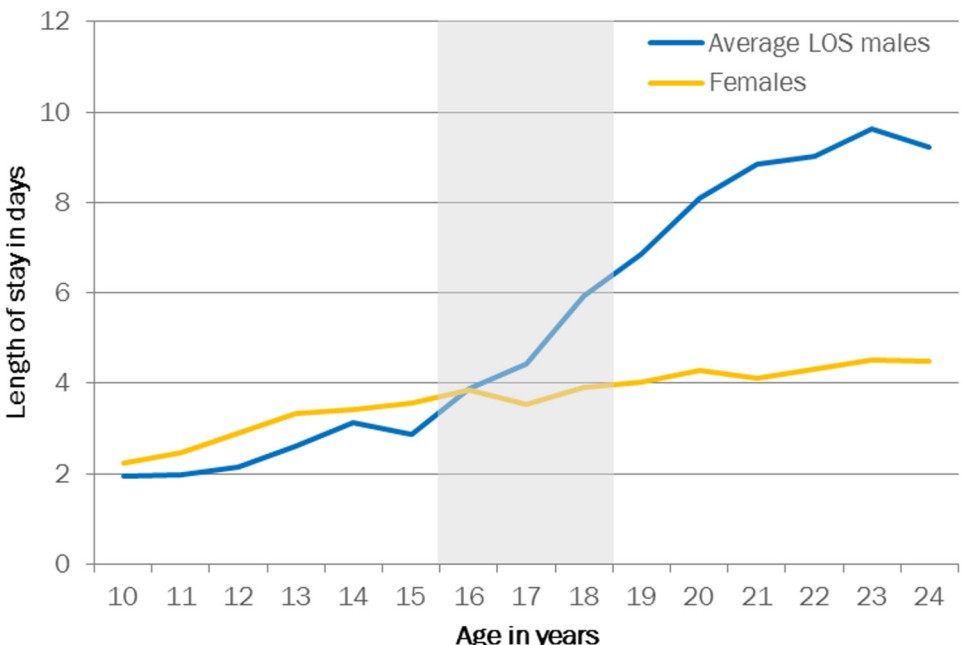

**Figure 5** Average length of stay (LOS) per admission by patient age. Grey area indicates age range for transition.

## DISCUSSION

Emergency admission rates increased at the age when young people transition from paediatric to adult health-care. The increase in emergency admission rates after the age of transition was significantly higher for those with underlying LTCs, compared with CYP without LTCs. Although emergency admissions increased in CYP with underlying LTCs (most steeply for mental health problems, present in 31% of included emergency admissions), the primary diagnosis on admission was often for other problems, most commonly signs and symptoms, and infections. Increases were more marked for girls than for boys, and for the most deprived CYP.

The average length of stay also increased after transition, particularly for boys. A substantial minority of the emergency admissions with long lengths of stay (10+ days) were for mental health conditions: schizophrenia and psychoses for boys, and eating and personality disorders for girls.

### Context

Our results are similar to the findings from a study in Ontario, Canada, that found an increase in the rates of emergency admissions for young people with diabetes during the transition from paediatric to adult services.[12] The rates for diabetes mellitus-related admissions were lower in our population (1–3 per 1000 vs 7–10 per 100 patient-years in Ontario) possibly due to differences in the population and in how conditions are coded. However, we found similar trends with regard to deprivation and female gender.

We found that while emergency admission rates for CYP with LTC increased across transition for most subgroups, the rates remained stable for CYP with cardiovascular disorders and cancer or blood disorders. This finding may reflect the intensity of paediatric and adult specialist care for these conditions and a reduced likelihood of emergency admission because other types of hospital care (eg, clinic review or planned admission) can be readily accessed. Alternatively, transition from paediatric services may be deferred.[24 25] For example, paediatric services for cancer can be commissioned up to the age of 24 years. A further explanation could be improved transition; for example, congenital heart disease services have reported using transition nurses to provide continuity between paediatric and adult services.[18 25]

Emergency admission rates were more likely to increase in children from more deprived backgrounds, although the effect size was small. We found that after transition emergency admission rates stabilised, both overall and for many LTC groups by the age of 24 years. Also taking the drop in the proportion of day cases and increase in length of stay into account, this could be explained by a higher admission threshold for adults and potentially unmet health needs as a result.

### Strengths and limitations

By analysing all NHS admissions in England, we avoided bias due to different referral pathways for CYP as they transition to adult services. Our analysis of underlying LTCs, based on any diagnostic codes recorded at the current or past admissions in the previous 5 years, showed that CYP with LTCs often present with primary diagnoses related to symptoms or conditions other than their LTC. Previous studies have focused on well-recognised LTCs such as asthma, diabetes, epilepsy and IBD,[26] but in our analyses these conditions accounted for only half of the increase in admissions and a minority of primary diagnoses. Our finding of relative stability in admissions for cancer and cardiovascular conditions suggests that care over the transition differs between conditions.

The cross-sectional study design of our study is a limitation. As we analysed admissions without taking into account that individuals could have multiple admissions (ie, clustering), we might have overestimated the contribution of certain conditions that are associated with multiple emergency admissions. Additionally, we may have included young people who were first diagnosed with LTCs after the age of transition.

We used a retrospective capture of past admissions to categorise CYP's emergency admissions into underlying LTC or no LTC. Our cross-sectional analyses of the whole population complement longitudinal analyses that follow individual CYP with and without LTCs across the period of transition. However, longitudinal analyses require identification of inception points, such as first diagnosis of the LTC. This is challenging using hospitalisation data as many CYP with underlying LTCs such as asthma are mostly managed in primary care. Such analyses would be feasible in future studies using linked primary care and hospitalisation data for whole populations.

### Implications

Emergency admissions may be a manifestation of the widely recognised difficulties young people face when they transition to adult health services.[1 18 27] Emergency admissions are more common and longer after transition, resulting in higher healthcare costs. Our findings of disparities according to the presence of underlying LTC and deprivation suggest potential to reduce emergency admissions across transition by focusing on young people with underlying LTCs. This will require system-level strategies and targeted investments.[28]

The high rates of emergency admissions for symptoms among CYP with LTCs may reflect a common preoccupation among adolescents with 'normal' physical symptoms, which generates more clinical concern and need for investigation in CYP with LTCs.[29 30] Adult care often involves less routine review or anticipatory management of problems and has been reported to be more focused on disease management rather than the CYP's broader health and psychosocial needs.[18] Alternatively, symptoms

may reflect undiagnosed complications or morbidity, or may be a manifestation of mental health distress, which might not be investigated in the context of an emergency hospital admission.

A potential unmet need for mental healthcare is borne out by the steep increase in emergency admissions with underlying mental health problems across transition. Previous research has claimed that UK mental health services around transition are not fit for purpose[31] and has called for prioritisation of initiatives and research to improve transition care.

Primary care is the first-line provider of mental healthcare for adolescents and the main provider for adults in England and elsewhere. However, rates of GP consultations decline steeply during adolescence, particularly among boys,[32] also reflected in our study by the decreasing proportion of admissions in older age groups that were referred by the GP. To improve provision of care during transition, services in primary and community care need to be made more attractive and accessible to adolescents to enable improved (mental) healthcare and thereby reduce the risk of admissions.[33–35] However, although there have been calls to improve primary care involvement in transition care, there is little evidence on the most effective ways to do this.[36] Further research into regional variation could help identify existing intervention or programmes with better outcomes for young people transitioning to adult care.

Hospital specialties in the UK are also making changes to transitional care. Several practice guidelines focus on early planning and communication, and having a single named practitioner to coordinate transition care and support.[4 18 27 37] Our findings suggest that future healthcare planning needs to shift from management of transition for specific LTCs to transitional care for the whole person that meets their holistic care needs, particularly mental health, thereby averting serious acute problems that require admission.[38]

**Acknowledgements** We would like to thank Jenny Woodman and David Cottrell for their input into the writing and amending of this manuscript, as well as the members of the Policy Research Unit in the Health of Children, Young People and Families: Catherine Law, Russell Viner, Miranda Wolpert, Amanda Edwards, Steve Morris, Helen Roberts, Terence Stephenson and Cathy Street.

**Contributors** LPMMW conceptualised and designed the study, carried out the analyses, drafted the initial manuscript, and approved the final manuscript as submitted. PH helped design the study, reviewed and revised the manuscript, and approved the final manuscript as submitted. AG helped conceptualise the study, critically reviewed the manuscript and approved the final manuscript as submitted. RG helped conceptualise and design the study, reviewed and revised the manuscript, and approved the final manuscript as submitted. All authors approved the final manuscript as submitted and agree to be accountable for all aspects of the work.

**Funding** LPMMW is funded by the Department of Health Policy Research Programme through funding to the Policy Research Unit in the Health of Children, Young People and Families (grant reference number 109/0001). PH is funded by a National Institute for Health Research postdoctoral fellowship (number PDF-2013-06-004). This article represents independent research funded by the National Institute for Health Research (NIHR) and the Department of Health. The views expressed in this publication are those of the author(s) and not necessarily those of the NHS, the National Institute for Health Research or the Department of Health. AG is funded by a Canadian Institutes for Health Research Applied Chair Award in Reproductive and Child Health Services Research. We also acknowledge the support from the Farr Institute of Health Informatics Research (MRC grant no: London MR/ K006584/1).

**Competing interests** None declared.

**Patient consent** Not required.

**Ethics approval** The use of Hospital Episodes Statistics data was approved by the Health and Social Care Information Centre for the purpose of this study (DARS-NIC-393510-D6H1D-v1.11). Source data can be accessed by researchers applying to the Health and Social Care Information Centre for England. Copyright 2017. Reused with the permission of the Health and Social Care Information Centre. All rights reserved.

**Provenance and peer review** Not commissioned; externally peer reviewed.

**Data sharing statement** Technical appendix with details on the code list used to identify primary diagnoses group is available in online supplementary appendix A. Statistical code is available upon request from authors.

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
