## [Reviewer comments · BMJ Open]

ARTICLE DETAILS

TITLE (PROVISIONAL)	Emergency admissions and long-term conditions during transition from paediatric to adult care: a cross-sectional study using Hospital Episode Statistics data
AUTHORS	Wijlaars, Linda; Hardelid, Pia; Guttman, Astrid; Gilbert, Ruth

VERSION 1 – REVIEW

REVIEWER	Jack Rusley Brown University, USA
REVIEW RETURNED	01-Feb-2018

GENERAL COMMENTS	Very well done and important study - thank you for your contribution to this field! No mention of IRB approval (or waiver). While there is discussion of data being de-identified, and approval from Health and Social Care Information Centre, because I am not familiar with this organization, it would be useful to provide more detail about whether a institutional review board or ethics board has reviewed and approved the study (perhaps the HSCIC is this body, but it's not clear from the statement in the manuscript).
---

REVIEWER	LARBRE Jean Paul Rheumatology Department, Hôpital Lyon Sud, 69310 Pierre Benite, France
REVIEW RETURNED	05-Feb-2018

GENERAL COMMENTS	The subject is of great interest. It is a large study. The results are interesting and well presented
--

REVIEWER	Alison White King's College London, UK
REVIEW RETURNED	10-Mar-2018

GENERAL COMMENTS	This is a well written and well designed study which contributes to existing knowledge. The study findings provide the basis for the development of longitudinal studies to understand the mechanisms which precipitate emergency admissions in different CYP groups so that interventions may be tested to meet their needs without recourse to an avoidable hospital admission.
--

REVIEWER	Stuart B. Bauer Boston Children's Hospital, 300 Longwood Avenue, Boston, Massachusetts 02115, USA
-----------------	--

REVIEW RETURNED	19-Mar-2018
-------------

GENERAL COMMENTS	This manuscript looking at admission rates for children and young adults before and after the ages of transition is a very thoughtful communication. The authors have done a considerable amount of work to see what the trends are, comparing those with and without long-term conditions. There are concerns and comments I feel the authors need to address to improve the quality and clarity of their submission.  1. In the Methods section the authors state this study involved patients admitted to an NHS Hospital over a 3-year time span, from April 2009 to March 2012. Can the authors explain why they chose this specific interval and why only over a 3 year span of time? Would a longer time span of assessment provided more information? Did anything occur during the time period chosen that influenced your decision to use the time period you did? 2. In the Methods section please expand why cross-section analysis was used as opposed to longitudinal follow-up in the same population. I know this would increase the time period for observation and delay publication several years but these patients could serve as their own controls. 3. Could the change in the numbers of admissions be related to a change in the denominator of patients from which each group of patients was drawn from? Is this a factor the authors looked into and if not why not? 4. Did the authors look at specific regions throughout Great Britain or whether there could be differences based on the type of Hospital? Sometimes, teaching or university hospitals may better prepare patients for transition than more a local or community hospital. The authors may not have been able to look at specific hospital groups give the hospital database they used. 5. Was there any change in the way patients with LTC were managed regarding transition in the last 3 -4 years versus the previous 3-4 years, as the increased awareness of the need to properly transition patients may have influenced the younger group versus the now older group (when they WERE at the younger age). 6. Table A1 has some discrepancies in numbers when I add up the figures involved in the 'No LTC ' and 'Overall' for the section "IMD '04 adding up rows 1 through 5. For those 2 sections I get 409,844 for 'No LTC' and 1,109,842 for 'Overall', when these totals should be 410,144 and 1,109,978, respectively. Please review and make corrections. 7. Several tables and appendices are not cited in the text and they should be (Table B1, C1, & C2; Appendix B). Some of the tables (C1 & C2) & Appendices (Appendices B & C) may not be necessary. Can the authors justify why they should be part of the manuscript?
--

VERSION 1 – AUTHOR RESPONSE

Reviewer 1	Response
No mention of IRB approval (or waiver). While there is discussion of data being de-identified, and approval from Health and Social Care Information Centre, because I	The study used non-identifiable information from the Health and Social Care Information Centre (HSCIC), and as such is exempt from UK National Research Ethics Committee approval as it involved secondary analysis of an existing dataset of anonymised data for service evaluation. Approvals for the use of the data were obtained from the data provider (HSCIC) as part of the

am not familiar with this organization, it would be useful to provide more detail about whether a institutional review board or ethics board has reviewed and approved the study (perhaps the HSCIC is this body, but it's not clear from the statement in the manuscript).	standard Hospital Episode Statistics approval process. We have clarified this by changing the heading for this approval from 'availability of data' to 'ethics approval' in the acknowledgments section (page 16)
Reviewer 2	
No comments	
Reviewer 3	
No comments	
Reviewer 4	
1. In the Methods section the authors state this study involved patients admitted to an NHS Hospital over a 3-year time span, from April 2009 to March 2012. Can the authors explain why they chose this specific interval and why only over a 3 year span of time? Would a longer time span of assessment provided more information? Did anything occur during the time period chosen that influenced your decision to use the time period you did?	The years of data used in the study were the most recent years of data available at the time of study inception. We used data covering three years to ensure any patterns were not due to single year fluctuations. We have added this to the methods section (page 5).
2. In the Methods section please expand why cross-section analysis was used as opposed to longitudinal follow-up in the same population. I know this would increase the time period for observation and delay publication several years but these patients could serve as their own controls.	We have added a sentence under the 'statistical analysis' heading expanding why we opted for a cross-sectional analysis (page 6): "As this is an exploratory study, we used a cross-sectional design." As detailed in the discussion (page 13), we agree with the reviewer that the longitudinal analysis, tracking the same individuals over time, would be of interest for future studies. However, in order to implement a longitudinal study design, it is necessary to define inception points for individual patients, which

	is beyond the scope of the current study.
3. Could the change in the numbers of admissions be related to a change in the denominator of patients from which each group of patients was drawn from? Is this a factor the authors looked into and if not why not?	The reviewer raises an important concern. We have aimed to account for changes in the source population by using admissions rates, rather than number of admissions. The denominator population for these rates are based on Mid-year population estimates from the Office of National Statistics (as detailed in the methods section on page 6). As a result, our findings should not be related to changes in the source population as we have accounted for this.
4. Did the authors look at specific regions throughout Great Britain or whether there could be differences based on the type of Hospital? Sometimes, teaching or university hospitals may better prepare patients for transition than more a local or community hospital. The authors may not have been able to look at specific hospital groups give the hospital database they used.	We agree with the reviewer that regional variation or variation by hospital type would be interesting perspectives to take into account. However as this is an exploratory study, using only cross-sectional data, it would be beyond our scope. We have included this as a suggestion for further research in our discussion: "Further research into regional variation could help identify existing intervention or programmes with better outcomes for young people transitioning to adult care" (page 15).
5. Was there any change in the way patients with LTC were managed regarding transition in the last 3 -4 years versus the previous 3-4 years, as the increased awareness of the need to properly transition patients may have influenced the younger group versus the now older group (when they WERE at the younger age).	The reviewer raises an important concern. As far as we are aware, there were no changes in national policies regarding management of children with long-term conditions relating to care around transition during this period. However, it is possible there were regional changes. However, as detailed in our response to question 3, assessing this would be beyond the scope of the present study.
6. Table A1 has some discrepancies in numbers when I add up the figures involved in the 'No LTC ' and 'Overall' for the section "IMD '04 adding up rows 1 through 5. For those 2 sections I get 409,844 for 'No LTC' and 1,109,842 for 'Overall', when these totals should be 410,144 and 1,109978, respectively. Please review and make	We thank the reviewer for noticing this error and have amended the number of children with missing IMD04 in the overall column (was 17,373, is now 17,409). However, the total in the IMD04 column for children with no LTC does add up to 410,144 for all subgroups.

corrections.	
7. Several tables and appendices are not cited in the text and they should be (Table B1, C1, & C2; Appendix B). Some of the tables (C1 & C2) & Appendices (Appendices B & C) may not be necessary. Can the authors justify why they should be part of the manuscript?	We thank the reviewer for thoroughly checking our appendices. We have referenced appendix B (including the table) in the methods section on page 6 and appendix C in the results section on page 10. Previously, we had not specifically called out the tables included in the appendices in our references in the text, but we have now included these.
Formatting amendments	
You have cited APPENDIX A right after APPENDIX B which makes your citations incorrect. Please review again your main document and ensure that all references will be cited and will appear in ascending order.	We have amended our appendices and changed appendix A to B and vice versa.

VERSION 2 – REVIEW

REVIEWER	Stuart B Bauer Boston Children's Hospital, Boston, MA, USA
REVIEW RETURNED	14-May-2018
GENERAL COMMENTS	The authors have satisfactorily answered all the concerns I raised and modified their manuscript accordingly.